

# New species of the endemic Neotropical caddisfly genus *Contulma* from the Andes of Ecuador (Trichoptera: Anomalopsychidae)

Ralph W. Holzenthal[1], Blanca Ríos-Touma[2] and Ernesto Rázuri-Gonzales[1,3]

[1] Department of Entomology, University of Minnesota, St. Paul, MN, United States of America
[2] Facultad de Ingenierías y Ciencias Agropecuarias. Ingeniería Ambiental; Grupo de Investigación en Biodiversidad, Medio Ambiente y Salud -BIOMAS-, Universidad de las Américas, Quito, Ecuador
[3] Departamento de Entomología, Museo de Historia Natural, Universidad Nacional Mayor de San Marcos, Lima, Peru

## ABSTRACT

The genus *Contulma* Flint (Trichoptera: Anomalopsychidae) is composed mostly of regionally endemic species occurring above 2,000 m, with a few more widespread species and some that are found at lower elevations. Adults of three new species of *Contulma* are described and illustrated from the Andes of Ecuador, *Contulma lina*, new species, *Contulma quito*, new species, and *Contulma sangay*, new species. These species are similar to previously described species from the region, including *C. paluguillensis, C. nevada*, and *C. lancelolata.* New provincial records are provided for *C. bacula, C. cataracta*, and *C. echinata. Contulma duffi Oláh, 2016* is considered a junior, subjective synonym of *C. penai, Holzenthal & Flint, 1995*. Also, we provide an identification key to males of the 30 *Contulma* species now known.

# INTRODUCTION

Neotropical Trichoptera currently includes more than 3,200 described species representing 155 genera and 25 families (*Holzenthal & Calor, 2017*) occurring in Mexico, the Caribbean, Central, and South America. Remarkably, 115 genera, or ca.75% of the total, are endemic to the region, making the fauna the second most diverse in the world for endemic genera after the Australasian (*De Moor & Ivanov, 2008*). Moreover, several Neotropical genera are highly endemic regionally at the species level such as *Amphoropsyche* Holzenthal (*Holzenthal, 1985*; *Holzenthal & Rázuri-Gonzales, 2011*), *Atananolica* Mosely (*Holzenthal, 1988*; *Henriques-Oliveira & Santos, 2014*), and *Contulma* Flint (*Holzenthal & Robertson, 2006*), the latter the main subject of this paper. Nevertheless, the Neotropical caddisfly fauna is incompletely known, mainly because there are regions in the Neotropics where the aquatic ecosystems are far from being well studied (e.g., *Ríos-Touma et al., 2017*). There is also a lack of regional researchers studying the order, especially in the Andean countries of Venezuela, Colombia, Ecuador, Peru, and Bolivia, which undoubtedly harbor hundreds

Corresponding author
Ralph W. Holzenthal,
holze001@umn.edu

**Table 1** Described species in the genus *Contulma* with geographic distribution and known life stages.

| Species | Author | Distribution | Known stages |
|---|---|---|---|
| *Contulma adamsae* | *Holzenthal & Flint (1995)* | Peru | male, female |
| *Contulma bacula* | *Holzenthal & Flint (1995)* | Colombia, Ecuador | male |
| *Contulma boliviensis* | *Holzenthal & Robertson (2006)* | Bolivia | male |
| *Contulma caldensis* | *Holzenthal & Flint (1995)* | Colombia | male |
| *Contulma cataracta* | *Holzenthal & Flint (1995)* | Ecuador | male |
| *Contulma colombiensis* | *Holzenthal & Flint (1995)* | Colombia | male, female |
| *Contulma costaricensis* | *Holzenthal & Flint (1995)* | Costa Rica | male |
| *Contulma cranifer* | *Flint (1969)* | Chile | male, female |
| *Contulma echinata* | *Holzenthal & Flint (1995)* | Colombia | male, female |
| *Contulma ecuadorensis* | *Holzenthal & Flint (1995)* | Ecuador | male, female |
| *Contulma fluminensis* | *Holzenthal & Robertson (2006)* | Brazil | male |
| *Contulma inornata* | *Holzenthal & Flint (1995)* | Colombia | male |
| *Contulma lanceolata* | *Holzenthal & Flint (1995)* | Ecuador | male |
| *Contulma lina*, **n. sp.** | | Ecuador | |
| *Contulma meloi* | *Holzenthal & Robertson (2006)* | Brazil | male |
| *Contulma nevada* | *Holzenthal & Flint (1995)* | Colombia | male, female, larva |
| *Contulma paluguillensis* | *Holzenthal & Ríos-Touma (2012)* | Ecuador | male, female, larva, pupa |
| *Contulma papallacta* | *Holzenthal & Flint (1995)* | Ecuador | male |
| *Contulma penai* | *Holzenthal & Flint (1995)* | Colombia, Ecuador | male, female, larva |
| *Contulma duffi* *Oláh, 2016*, **n. syn.** | | | male |
| *Contulma quito*, **n. sp.** | | Ecuador | |
| *Contulma sana* | *Jardim & Nessimian (2011)* | Brazil | male |
| *Contulma sancta* | *Holzenthal & Flint (1995)* | Costa Rica | male |
| *Contulma sangay*, **n. sp.** | | Ecuador | |
| *Contulma spinosa* | *Holzenthal & Flint (1995)* | Colombia, Ecuador | male, female, larva |
| *Contulma talamanca* | *Holzenthal & Flint (1995)* | Costa Rica | male, female |
| *Contulma tapanti* | *Holzenthal & Flint (1995)* | Costa Rica | male, female |
| *Contulma tica* | *Holzenthal & Flint (1995)* | Costa Rica | male |
| *Contulma tijuca* | *Holzenthal & Flint (1995)* | Brazil | male, female, probable larva |
| *Contulma tripui* | *Holzenthal & Robertson (2006)* | Brazil | male |
| *Contulma valverdei* | *Holzenthal & Flint (1995)* | Costa Rica | male, female, larva |

of species. For example, co-authors Ríos-Touma is the first Ecuadorian to describe new species of caddisflies and Rázuri-Gonzales only the third Peruvian to do so.

A comprehensive revision of the endemic Neotropical genus *Contulma* was completed by *Holzenthal & Flint (1995)* and included 21 species, 18 described as new. Since then, seven new species have been described, including one we synonymize here (Table 1). Species in the genus are known from Costa Rica, the Andes of Colombia to Chile, and in the mountains of southeastern Brazil (*Holzenthal & Calor, 2017*). This genus seems to display a high degree of local endemism among its species (*Holzenthal & Robertson, 2006*), and they are rarely collected using standard light trap techniques. Hand netting during the day, especially at high elevations, or the use of Malaise traps is generally more effective

(*Holzenthal & Ríos-Touma, 2012*). The infrequency of collection does not mean the species are rare in nature, but most have been described from only 1–5 individuals. Perhaps this is a reflection of minimal collecting effort or low temperatures at high elevations that reduce adult flying activity. The habitats of these species are small waterfalls, seeps, and small streams in lush forested mountainous areas as well as high elevation *páramo* streams above the tree line in the Andes (*Holzenthal & Flint, 1995*).

Of the 30 species now known in the genus, including three new species described here, 19 occur in the tropical Andean countries (Ecuador, Colombia, Peru, Bolivia) and all occur above 2,000 m, except for two of these species also found in lower elevations (*Holzenthal & Flint, 1995*; *Holzenthal & Robertson, 2006*). Eight species occur in Ecuador, five endemic and three are also present in Colombia. The three new species described here are from localities were no caddisfly collecting occurred previously.

## MATERIALS AND METHODS

We used the methods described by *Blahnik & Holzenthal (2004)* to prepare adult specimens for taxonomic study. Genitalia were cleared in 85% lactic acid heated to 125 °C for 20 min (*Blahnik, Holzenthal & Prather, 2007*). An Olympus BX41 compound microscope outfitted with a drawing tube was used to observe specimens. Genital structures were drawn with pencil on paper and final illustrations were rendered in Adobe Illustrator. Morphological terminology follows that of *Holzenthal & Flint (1995)*. Descriptions of species and generation of the identification key were accomplished using the software packages DELTA and INTKEY (*Dallwitz, 1980*; *Dallwitz, Paine & Zurcher, 1999*).

Types of the new species are deposited in the collections of the Museo Ecuatoriano de Ciencias Naturales, Quito, Ecuador (MECN) and the University of Minnesota Insect Collection, St. Paul, Minnesota, USA (UMSP). All specimens examined in this study were affixed with a barcode label containing a unique nine digit numeric code starting with the prefix UMSP. These codes are provided here for holotypes only. All associated specimen data are stored in the UMSP database. This study was performed under the Environmental Ministry of Ecuador study permits 36-2010-IC-FLO/FAU-DPA-MA and 005-15-IC-FAU-FLO-DNB/MA.

The electronic version of this article in Portable Document Format (PDF) will represent a published work according to the International Commission on Zoological Nomenclature (ICZN), and hence the new names contained in the electronic version are effectively published under that Code from the electronic edition alone. This published work and the nomenclatural acts it contains have been registered in ZooBank, the online registration system for the ICZN. The ZooBank LSIDs (Life Science Identifiers) can be resolved and the associated information viewed through any standard web browser by appending the LSID to the prefix http://zoobank.org/. The LSID for this publication is: urn:lsid:zoobank.org:pub:54BC56DC-5CC1-4DA0-82E4-AF69599C2F5D. The online version of this work is archived and available from the following digital repositories: PeerJ, PubMed Central and CLOCKSS.

## RESULTS

## Species descriptions

***Contulma lina*, new species, Holzenthal, Ríos-Touma, Rázuri-Gonzales**
LSID urn:lsid:zoobank.org:act:310DDC54-0008-4535-A385-BD71BB630F21
Figs. 1A–1E, 2

**Diagnosis:** *Contulma lina*, n. sp., as well as *C. quito*, n. sp., and the recently described *C. paluguillensis Holzenthal & Ríos-Touma, 2012*, in addition to *C. echinata*, *C. nevada*, and *C. papallacta*, all described by *Holzenthal & Flint (1995)*, and all from either Ecuador or Colombia, share several features. All have some degree of development of a dorsolateral process on segment IX in the male genitalia and a development of setae or a setose process on the mesal face of the same segment below the dorsolateral process. These three species formed a clade within the *cranifer*-group of *Holzenthal & Flint (1995)* to which *C. lina*, n. sp., *C. quito*, n. sp., and *C. paluguillensis* also belong. In some species, one or the other of these characters may be more or less developed, but the combination of characters shared by these species indicate that they may have a common origin. The phallic structures seem to be unique to each species and attest to their distinctiveness. All are known from only a handful of specimens, most from only the type and a few paratypes, and the species each occur from only one or a few high altitude localities spread across a vast expanse of the northern Andes. We predict that additional collecting will discover yet more new species in this radiation of Andean Trichoptera biodiversity. *Contulma lina* is distinguished from the above mentioned species, all illustrated by *Holzenthal & Flint (1995)* and *Holzenthal & Ríos-Touma (2012)*, by the following combination of characters: mesal surface of segment IX bearing a sinuous band of setae from below dorsolateral process and continuing to near sternum IX; mesal process of sternum IX with prominent posteromesal, heavily sclerotized, spatulate projection, without excavation; and phallus without spine-like setae, but instead with the apex bearing a lightly sclerotized scale-like structure, shallowly excavate at apex.

**Description:** *Male*: Forewing length 5.5 mm ($n = 1$). Forewing color gray brown, immaculate, vestiture rubbed (specimen was netted during the day in light rain and mist). Male genitalia: Segment IX very short dorsally, narrow, deeply excavate mesally; in lateral view, IX quadrate, extended anterolaterally; posteriorly with short, broad, dorsolateral, spatulate process; posterior margin of IX produced medially to form broad, prominent, quadrate, heavily setose, paired lateral lobes; lobes widely separated ventrally; mesal face of segment IX bearing sinuous band of setae from below spatulate process continuing to near sternum IX; sternum IX with prominent posteromesal, heavily sclerotized, spatulate projection. Inferior appendages short, subtriangular, apices broadly rounded and bearing apical setae; inferior appendages apparently fused to base of IXth sternal projection, together forming highly complex structure as in Figs. 1A, 1C. Processes of subphallic membranes present, membranous, mound-like, setose. Segment X entirely membranous, apex entire, extending beyond apices of dorsolateral processes. Phallus complex; phallobase tubular, elongate, slender, sclerotized; phallicata very lightly sclerotized, dorsally with paired, very lightly sclerotized, semi-membranous lobes; apicoventral phallic membranes with paired,

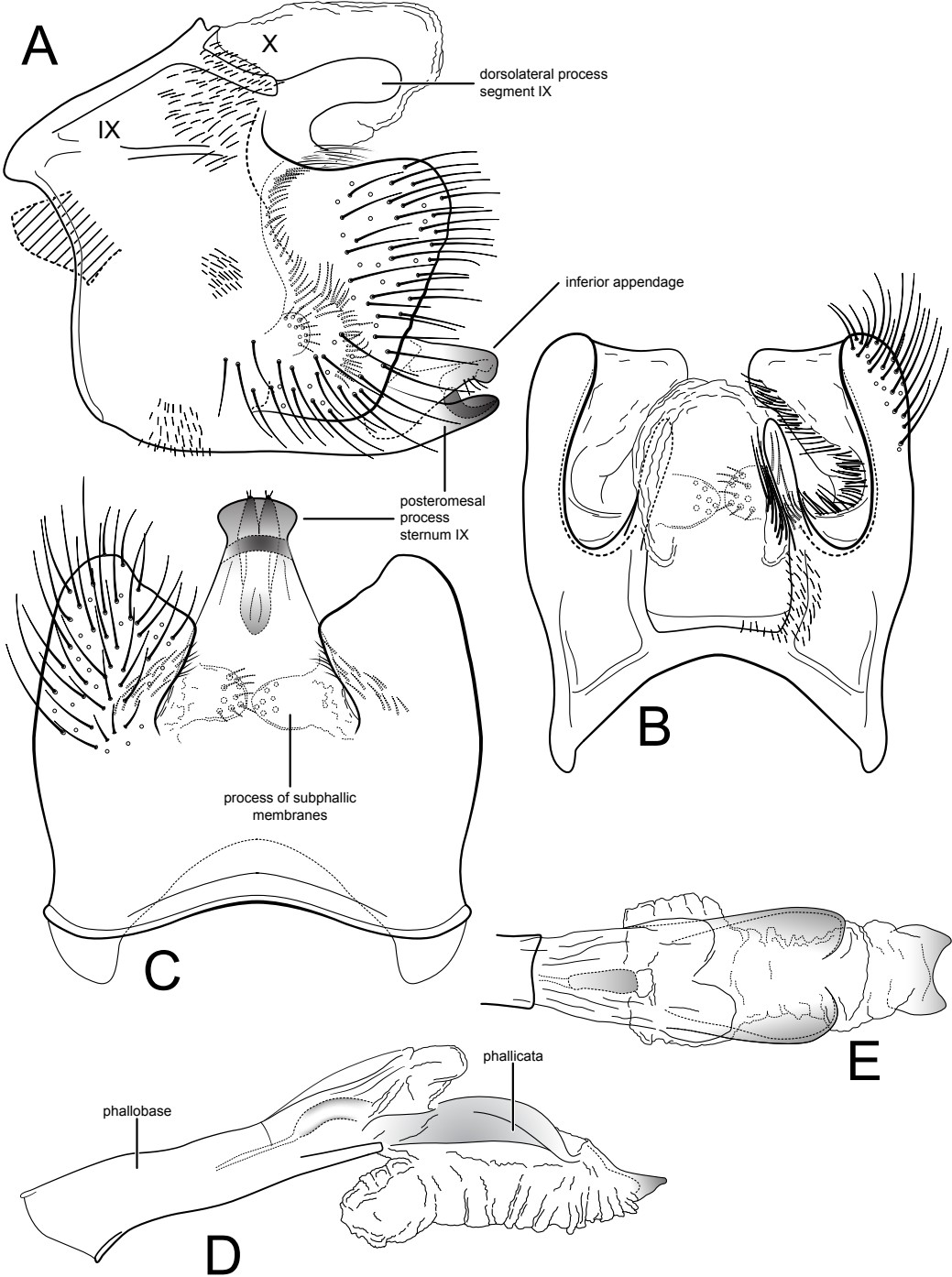

**Figure 1** ***Contulma lina*, new species, male genitalia.** Male genitalia of *Contulma lina*, new species. (A) segments IX and X, lateral (base of phallus indicated in crosshatch). (B) segments IX and X, dorsal. (C) segment IX, ventral. (D) phallus, lateral. (E) phallus apex, dorsal. Abbreviations: IX, abdominal segment IX; X, abdominal segment X.

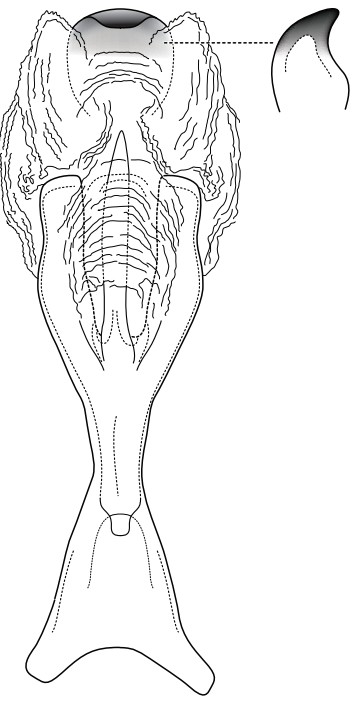

**Figure 2** *Contulma lina* **new species, female vaginal apparatus.** Female vaginal apparatus of *Contulma lina*, new species, ventral.

broad, lightly sclerotized, dorsolateral plates, membranes inflated anteroventrally, apically with lightly sclerotized scale-like structure, shallowly excavate at apex; phallotremal sclerite present, tubular, curved.

*Female*: Forewing length 6.5 mm ($n = 1$). Color and vestiture as in male. Vaginal apparatus in ventral view elongate, hourglass shaped, narrowest in middle; base widely emarginate, subtriangular; apex trident shaped with wide, paired, sclerotized, slightly sinuate midlateral processes, their apices truncate; single medial process elongate, narrow, slerotized, about same length as midlateral processes; medial membranes highly convoluted, as approximated in Fig. 2; apical membranes highly convoluted, with heavily sclerotized, rounded, beak-like apical sclerite.

**Holotype male: ECUADOR: Napo:** Reserva Ecológica Cayambe-Coca, waterfall, rd. to Oyacachi, 0.32621°S, 78.15049°W, el. 3,690 m, 16.x.2011, Holzenthal, Ríos-Touma, Pita (UMSP000098532) (UMSP). **Paratype:** same data as holotype, one female (MECN).

**Etymology:** This species in named for Lina Pita, lifelong friend of Blanca Ríos-Touma and one of the collectors of this species.

*Contulma quito*, **new species, Holzenthal, Ríos-Touma, Rázuri-Gonzales**
LSID urn:lsid:zoobank.org:act:97969526-CF58-48A8-8548-8817326D04A0
Figs. 3A–3F

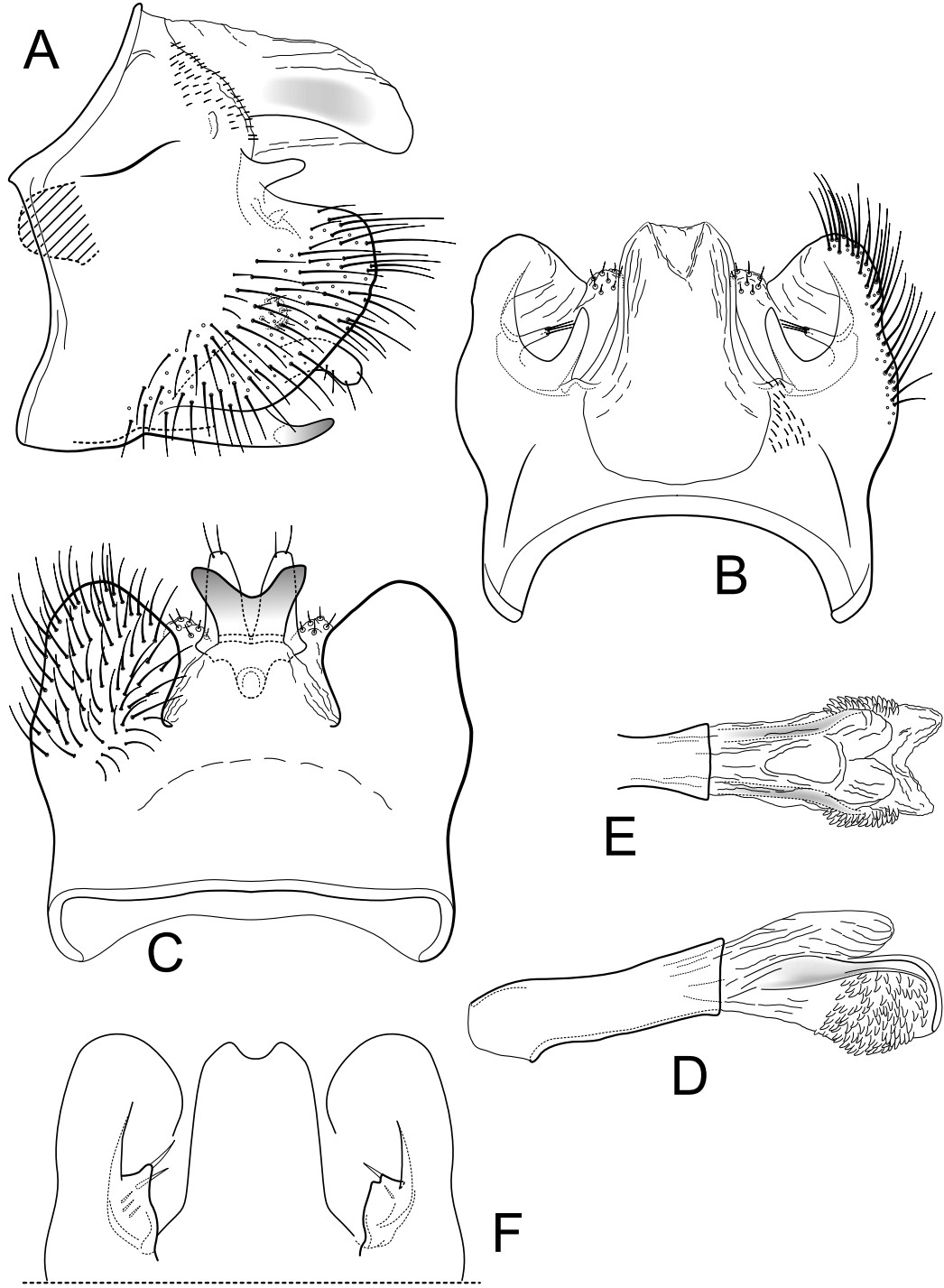

**Figure 3** *Contulma quito.* Male genitalia of *Contulma quito*, new species. (A) segments IX and X, lateral (base of phallus indicated in crosshatch). (B) segments XI and X, dorsal. (C) segment IX, ventral. (D) phallus, lateral. (E) phallus apex, dorsal. (F) segments IX and X, dorsal, variation in paratype specimen UMSP000148995.

**Diagnosis:** This new species is very similar to *C. nevada* of the group of species mentioned above (*C. echinata*, *C. lina*, *C. nevada*, *C. paluguillensis*, and *C. papallacta*). *Contulma quito*, n. sp., and *C. nevada* have similar short dorsolateral processes on segment IX, a small setal patch or setal bearing process below the dorsolateral process, emarginate sternal projection of sternum IX, and short spines in the phallus. The major differences between *C. quito* and *C. nevada* are the broader excavation of the IXth sternal projection and the many more spine-like setae in the phallus of *C. quito*.

**Description:** *Male*: Forewing length 5.5 mm ($n = 3$). Forewing color brown, with small patch of cream colored hairs at apex of subcosta and arculus, vestiture intact. Male genitalia: Segment IX very short dorsally, narrow, deeply excavate mesally; in lateral view, IX quadrate, slightly extended anterolaterally; posteriorly with very short, narrow, dorsolateral, spatulate process; posterior margin of IX produced medially to form broad, prominent, broadly rounded, heavily setose, paired lateral lobes; lobes widely separated ventrally; mesal face of segment IX with short, sclerotized, setose process below spatulate process. (*Variation:* in male paratype UMSP000148995, the spatulate processes and the short setose processes below them are each fused on both sides of the specimen to form a thin, crecentic shelf (Fig. 3F). In male paratype UMSP000148996, the spatulate processes and the setose processes below them are variously slightly longer or shorter on either the right or left sides than they are in the holotype). Sternum IX with prominent posteromesal, heavily sclerotized, strongly emarginate projection. Inferior appendages short, crescentric, apices rounded and bearing apical setae; inferior appendages apparently fused to base of IXth sternal projection, together forming highly complex structure as in Figs. 3A, 3C. Processes of subphallic membranes present, membranous, mound-like, setose. Segment X entirely membranous, apex divided, with lightly sclerotized lateral flanges. Phallus complex; phallobase tubular, elongate, slender, sclerotized; phallicata very lightly sclerotized, dorsally with paired, very lightly sclerotized, semi-membranous lobes; apicoventral phallic membranes with paired, broad, lightly sclerotized, dorsolateral plates forming broadly rounded trough, apicoventrally with paired patches of numerous (ca. 50) very short spine-like setae; phallotremal sclerite very lightly sclerotized, difficult to discern.

*Female*: Unknown.

**Holotype male: ECUADOR: Pichincha:** Distrito Metropolitano de Quito, Quebrada Guapalito, 0.40113°S, 78.38378°W, el. 2,807 m, 26.vii.2015, Rázuri, Ríos-Touma, Amigo (UMSP000148997) (UMSP). **Paratypes: ECUADOR: Pichincha:** Distrito Metropolitano de Quito, Quebrada Convalescencia, 0.40060°S, 78.38256°W, el. 2,813 m, 26.vii.2015, Morabowen, Hernández, two males (UMSP, MECN).

**Etymology:** Named for Quito, the capital of Ecuador, where the species was collected and where the surrounding Andes seem to harbor a multitude of species in the genus.

***Contulma sangay*, new species, Holzenthal, Ríos-Touma, Rázuri-Gonzales**
LSID urn:lsid:zoobank.org:act:38C17810-F31C-45A5-BFAC-3D3C5E68AD47
Figs. 4A–4D, 5

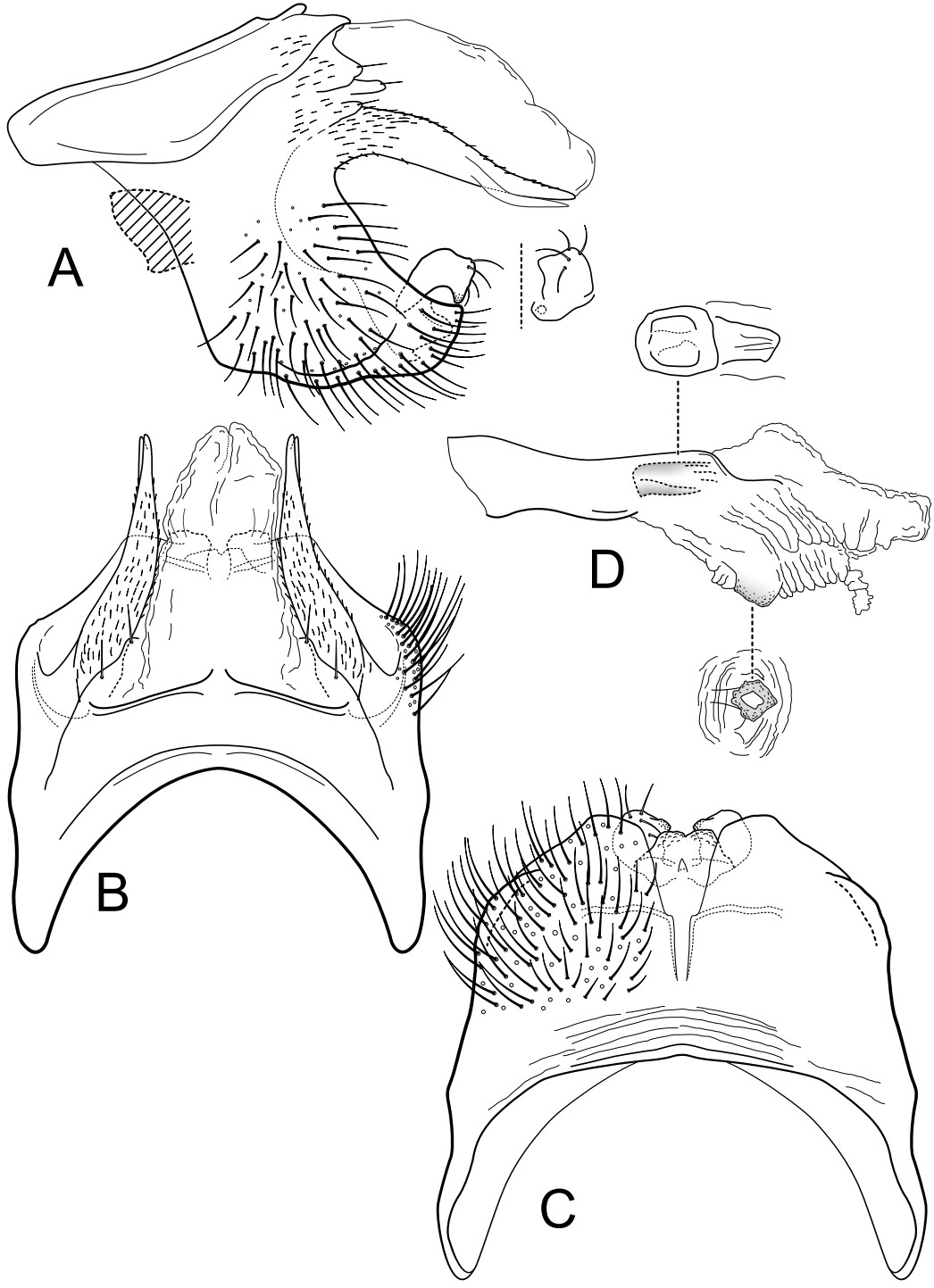

**Figure 4** *Contulma sangay,* **new species, male genitalia.** Male genitalia of *Contulma sangay*, new species. (A) segments IX and X, lateral (base of phallus indicated in crosshatch); inset: apex of inferior appendage, caudal. (B) segments XI and X, dorsal. (C) segment IX, ventral. (D) phallus, lateral; insets: details of phallic structures.

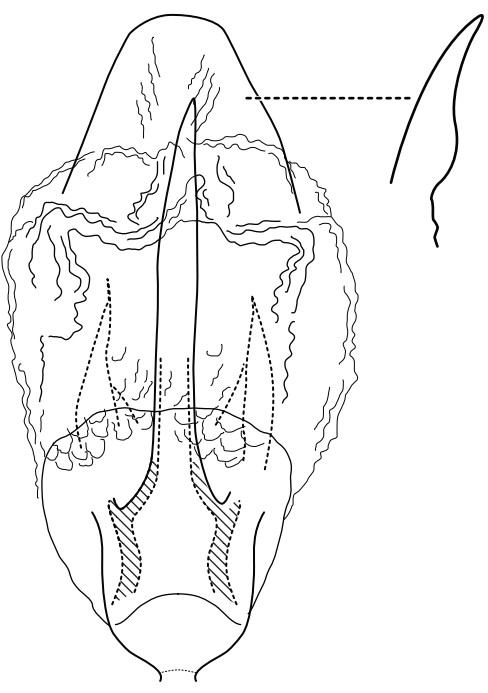

**Figure 5** ***Contulma sangay*, new species, female vaginal apparatus.** Female vaginal apparatus of *Contulma sangay*, new species, ventral.

**Diagnosis:** This new species is similar to *C. lanceolata*, *Holzenthal & Flint, 1995*, also from Ecuador. Both share a similar shape of segment IX, which is strongly extended anterolaterally and with the posterior portion excavated medially in both species. In addition, both have an elongate dorsolateral setose lanceolate process on segment IX, but in *C. sangay* the process is divided apically into a pair of narrow, terete lobes. Most distinctively, the new species lacks the ventrolateral process of segment IX seen in *C. lanceolata* (*Holzenthal & Flint, 1995*, fig. 70).

**Description:** *Male*: Forewing length 4.5 mm ($n = 1$). Forewing color brown, immaculate, vesture intact. Male genitalia: Segment IX short dorsally, narrow; in lateral view, IX trapezoidal, strongly extended anterolaterally; posteriorly with elongate, dorsolateral, lanceolate process, apically divided into pair of terete, narrow, closely appressed lobes; process bearing minute setae dorsally along length; posterior margin of IX excavated medially, produced ventrally, to form prominent, semiquadrate, setose, paired lateral lobes; lobes close together ventrally, forming acute separation; mesal face of segment IX without setae or processes; sternum IX short, subquadrate projection, its suface rugose and with very small, mesal sclerotized spur. Inferior appendages very short, crescentric, apices subacute, directed ventrally, and bearing apical setae; inferior appendages apparently fused to base of IXth sternal projection, together forming highly complex structure as in Figs. 4A, 4C. Processes of subphallic membranes absent (or not apparent). Segment X entirely membranous, apex cleft, extending to apices of dorsolateral processes. Phallus complex;

phallobase tubular, elongate, slender, sclerotized; phallicata very lightly sclerotized, dorsally with prominent membranous lobe; apicoventral phallic membranes with paired, broad, lightly sclerotized, ventrolateral plates, fused ventrally and forming apparent pore, apex with pair of small papillate lobes; phallotremal sclerite present, small, subquadrate.

*Female*: Forewing length 4.5 mm ($n = 1$). Color as in male, but vestiture rubbed. Vaginal apparatus in ventral view short, oval, widest in middle; base rounded, urn-shaped; apex trident shaped with narrow, paired, lightly sclerotized, indistinct midlateral processes, their apices acute; single medial process elongate, narrow, sclerotized, longer than midlateral processes; medial membranes highly convoluted, as approximated in Fig. 5; apical membranes highly convoluted, with lightly sclerotized, broad, thin, shelf-like apical sclerite.

**Holotype male: ECUADOR: Morona-Santiago:** Río Salado, Highway E46 (via Riobamba-Macas), 2.24253°S, 78.27791°W, el. 1,646 m, 26.i.2015, Holzenthal, Huisman, Ríos-Touma, Amigo (UMSP000147014) (UMSP). **Paratype:** same data as holotype, one female (MECN).

**Etymology:** Named for the type locality, Sangay National Park, where the species was collected from the Río Salado.

## New provincial records

***Contulma bacula*** *Holzenthal & Flint, 1995*:11 (Type locality: Ecuador, Napo, 1 mi E of Papallacta; type depository: NMNH; holotype male).—*Medellín, Ramírez, & Rincón, 2004*:201 (distribution; biology).—*Holzenthal & Calor, 2017*:21 (catalog).
**Distribution.** Colombia, Ecuador.
**New provincial record: ECUADOR: Morona-Santiago:** Río Tinguichaca; Highway E46 (via Riobamba-Macas), 02.21474°S, 078.44218°W, el. 2,772 m, 25.i.2015, Holzenthal, Huisman, Ríos-Touma, Amigo, one male, two females (UMSP).

***Contulma cataracta*** *Holzenthal & Flint, 1995*:12 (Type locality: Ecuador, Napo, Río Maspa Chico, 2 km W Cuyuja; type depository: NMNH; holotype male).—*Holzenthal & Calor, 2017*:22 (catalog).
**Distribution.** Ecuador.
**New provincial record: ECUADOR: Morona-Santiago:** Sangay National Park, waterfall 2, 2.18111°S, 78.5062°W, el. 3,516 m, 12.xi.2015, Ríos-Touma, Thomson, Amigo, three males, one female (UMSP).

***Contulma echinata*** *Holzenthal & Flint, 1995*:15 (Type locality: Colombia, Caldas, 5 km W Termales de Ruíz; type depository: NMNH; holotype male; female).—*Muñoz Quesada, 2000*:274 (checklist).—*Holzenthal & Calor, 2017*:22 (catalog).
**Distribution.** Colombia, Ecuador.
**New provincial record: ECUADOR: Napo:** Reserva Ecológica Cayambe-Coca, waterfall, rd. to Oyacachi, 0.32621°S, 78.15049°W, el. 3,690 m, 16.x.2011, Holzenthal, Ríos-Touma, Pita, one male (UMSP); same, except 26.ii.2012, Ríos-Touma and Pita, one male (UMSP).

## New synonymy

*Contulma penai* *Holzenthal & Flint, 1995*:18 (Type locality: Ecuador, Zamora-Chinchipe, 30 km E Loja; type depository: NMNH; holotype male; female; larva).—*Muñoz Quesada, 2000*:274 (checklist).—*Holzenthal & Calor, 2017*:23 (catalog).

—*Contulma duffi* *Oláh, 2016*: 169 (Type locality: Colombia, Antioquia, Dusky Starfrontlet Bird Reserve, Cordillera Occidental, Urrao, 6°25′N, 75°05′W, 9.ii.2014, caught by hand, leg. A.G. Duff; type depository: private collection of J. Oláh; holotype male) **NEW SYNONYM**

**Distribution.** Colombia, Ecuador.

*Contulma duffi*, described from Antioquia, Colombia, fits clearly within the variation we have seen in the species *C. penai*, which also occurs in Antioquia, Colombia, and Ecuador, where it is rather common compared to other species in the genus. There are no features illustrated or diagnosed in the original description that distinguish the species from *C. penai*. Most of the characters discussed by *Oláh (2016)* are those common to the genus as a whole. The membranous structure of tergum X is identical in both species. The dorsolateral processes of segment IX are illustrated as slightly curved in *C. duffi* compared to those illustrated for *C. penai*, but this slight difference is variable, and the inferior appendages are identical in the two species. In Oláh's diagnosis, the species is said to be most similar to *C. bacula*, but it shares little in common with that species. The inferior appendages, the posteromesal process of sternum IX, and the phallus are completely different between the 2 species.

## Key to males of *Contulma* Species[1]

[1]Citations of previously published illustrations referenced in the key are abbreviated as follows: HF, *Holzenthal & Flint, 1995*; HR, *Holzenthal & Robertson, 2006*; JN, *Jardim & Nessimian, 2011*; HRT, *Holzenthal & Ríos-Touma, 2012*.

1. Posterior margin of segment IX with dorsolateral processes (HF 24, 44, 58, 79, 90, 107) (these processes may be short, but present, as in *C. quito*, Figs. 3A, 3B) ................................................................................................................. 2

   Posterior margin of segment IX without dorsolateral processes, although there may be small setose projections or patches of setae (HF 40, 48, 49, 86, 95, 112) ................................................................................................................. 21

2(1). Dorsolateral processes of segment IX very long, slender, strongly down curved, their apices rugose (HF 107, HR 2A, 4A) ........................................................... 3

   Dorsolateral processes of segment IX shorter and/or differently shaped, their apices not rugose ................................................................................................. 6

3(2). Phallus with pair of large, highly membranous convoluted dorsolateral lobes (in some specimens these lobes may not evert during the clearing process) (HR 2D-F, 4D-E) ........................................................................................... 4

   Phallus without highly membranous convoluted lobes or with much smaller, less well-developed lobes (HF 110, JN 5) ........................................................... 5

4(3). Segment IX extended anterodorsally (HR 4A); setose lobe of posterior margin of segment IX situated close to middle of segment (HR 4A) ........................... ................................................................................................... *Contulma tripui*

Segment IX only slightly extended anterodorsally, if at all (HR 2A); setose lobe of posterior margin of segment IX situated close to ventral margin of segment (HR 2A) ................................................................... *Contulma fluminensis*

5(3). Posterior margin of segment IX extended into a long, narrow, acute, setose lobe (JN 1) ................................................................ *Contulma sana*

Posterior margin of segment IX extended into a much shorter, subtriangular, setose lobe (HF 107) ................................................................ *Contulma tijuca*

6(2). Phallus with pair of large, highly membranous convoluted dorsolateral lobes (in some specimens these lobes may not evert during the clearing process) (HF 28, 33, 56, 93) ................................................................................................. 7

Phallus without highly membranous convoluted lobes or with much smaller, less well-developed lobes (HF 81) ...................................................... 10

7(6). Segment IX with pair of elongate, heavily setose, ventrolateral lobes (HF 24, 26) ...................................................................................... *Contulma adamsae*

Segment IX without such lobes ...................................................................... 8

8(7). Dorsolateral processes of segment IX, in lateral view, long, linear, directed ventrad along entire length (sinuous and crossing apically in dorsal view) (HF 52–53); segment IX very short dorsolaterally (HF 52); segment X membranous, almost obliterated (HF 53) ...................................................... *Contulma cranifer*

Dorsolateral processes of segment IX, in lateral view, much shorter, linear to lorate, directed posteriad, apex only slightly to strongly directed ventrad (straight and not crossing apically in dorsal view); segment IX long dorsolaterally segment X membranous, but well developed (HF 30, 31, 90) .................. 9

9(8). Dorsolateral processes of segement IX, narrow, terete, apex slightly directed ventrad (HF 30–31) ................................................................ *Contulma bacula*

Dorsolateral processes of segment IX lorate (strap-shaped with apex flexed), apex strongly directed ventrad (HF 90) ............................... *Contulma spinosa*

10(6). Segment IX with mesolateral patch of long or short spine-like setae or with mesolateral setose protuberances (HF 57, 75, 79; HRT 1B; Figs. 1A, 1B, 3B) ... ............................................................................................................................... 11

Segment IX without mesolateral setae or setose protuberances ..................... 15

11(10). Apex of phallus with pair of large, tooth-like structures and/or with smaller apicoventral spines (HF 60, 81; Fig. 3D) ........................................................ 12

Apex of phallus without such spines, although small setae or papillae may be present ........................................................................................................... 14

12(11). Apex of phallus with pair of large, tooth-like structures and smaller apicoventral spines (HF 60) ............................................................. *Contulma echinata*

Apex of phallus with pair of large, tooth-like structures only (HF 81) ...............
.......................................................................................... *Contulma papallacta*

Apex of phallus with smaller apicoventral spines only (HF 77; Figs. 3D, 3E) .....
.................................................................................................................. 13

13(12). Projection of sternum IX subtriangular in ventral view, with broad, angulate mesal excavation (Fig. 3C) ................................... *Contulma quito,* **new species**

Projection of sternum IX quadrate in ventral view, with narrow mesal excavation (HF 76) ........................................................................... *Contulma nevada*

14(11). Mesolateral setae of segment IX forming sinuous band from below dorsolateral process continuing to near sternum IX (Figs. 1A, 1B); projection of sternum IX rounded (entire) apically (Fig. 1C) ........................ *Contulma lina,* **new species**

Mesolateral setae of segment IX restricted to patch below dorsolateral process (HRT 1A, B); projection of sternum IX emarginate apically (HRT 1C) ............
.................................................................................... *Contulma paluguillensis*

15(10). Segment IX extended anterodorsally (HF 44, 61, 70, 82; HR 1A; Fig. 4A); inferior appendage, in lateral view, crescentic ......................................................... 16

Segment IX only slightly extended anterodorsally, if at all (HF 66); inferior appendage, in lateral view, subtriangular or quadrate (HF 66) ..............................
.................................................................................................. *Contulma inornata*

16(15). Apex of phallus with pair of large, tooth-like structures (HF 64, 65); tergum IX with prominent, porsteromesal extension (HF 61, 62) ...................................
.............................................................................................. *Contulma ecuadorensis*

Apex of phallus without such spines, although small setae or papillae may be present; tergum IX without posteromesal extension ...................................... 17

17(16). Dorsolateral processes of segment IX clothed with short, fine setae (HF 44, 71, 82; Figs. 4A, 4B) ................................................................................... 18

Dorsolateral processes of segment IX lacking vestiture of fine setae (HR 1A, B)
.................................................................................................. *Contulma boliviensis*

18(17). Dorsolateral processes of segment IX curved mesally, widely separated, subtriangular (HF 44–46) ...................................................... *Contulma colombiensis*

Dorsolateral processes of segment IX straight, parallel, directed downward, close together, lanceolate ................................................................................. 19

19(18). Segment IX posteriorly with only single long dorsolateral or lateral setose process or none at all ...................................................................................... 20

Segment IX posteriorly with both long dorsolateral setose process and long, slender, sharply pointed, ventrolateral process (HF 70–72) ......................................................................................................................... ***Contulma lanceolata***

20(19). Apex of dorsolateral processes of segment IX entire (HF 82, 83) ............................................................................................................. ***Contulma penai***

Apex of dorsolateral processes of segment IX divided into pair of terete, narrow, closely appressed lobes (Figs 4A, 4B) ................................................................................................ ***Contulma sangay*, new species**

21(1). Segment X with lightly sclerotized lateral regions that apparently articulate basally with sclerotized projections of dorsolateral corners of segment IX (HF 48–49, 103–104) ................................................................... 22

Segment X without lightly sclerotized lateral regions that apparently articulate basally with sclerotized projections of dorsolateral corners of segment IX .. ................................................................................................................. 23

22(21). Segment IX with mesolateral patch of short spine-like setae (HF 48–49); projection of sternum IX truncate (HF 50); parameres long, slender (HF 51) ........ .......................................................................................... ***Contulma costaricensis***

Segment IX without mesolateral setae (HF 104); projection of sternum IX acute apically (HF105); parameres short (HF 106) ..................... ***Contulma tica***

23(21). Phallus with pair of large, highly membranous convoluted dorsolateral lobes (in some specimens these lobes may not evert during the clearing process) (HF 42–43, 89, 114–115) ................................................................................ 24

Phallus without highly membranous convoluted lobes or with much smaller, less well-developed lobes ............................................................................. 26

24(23). Segment IX with mesolateral patch of long (HF 39–40) or short (HF 86) spine-like setae ..................................................................................................... 25

Segment IX without mesolateral setae or setose protuberances (HF 111–112) .. .................................................................................................. ***Contulma valverdei***

25(24). Dorsolateral membranous lobes of phallus each ending in sclerotized, scale-like process (HF 42–43); projection of sternum IX long, rounded (entire) apically (HF 41) ............................................................... ***Contulma cataracta***

Dorsolateral membranous lobes of phallus entirely membranous (HF 89); projection of sternum IX short, slightly emarginate apically .................................... .................................................................................................. ***Contulma sancta***

26(23). Segment IX with mesolateral patch of long or shorter spine-like setae (HF 95, 100) or with mesolateral setose protuberances (HF 36); inferior appendage, in lateral view, crescentic ...................................................................................... 27

[2]*Contulma meloi* is a member of the group of species including *C. fluminensis*, *C. sana*, *C. tijuca*, and *C. tripui* from southeastern Brazil. However, it lacks the very long, slender, strongly downcurved, apically rugose dorsolateral processes of segment IX. The species is known only from the male holotype and a male paratype. In the holotype, there are no dorsolateral processes (HR 3A), but in the paratype, a rudimentary process occurs (HR 3G). However, it shares all other diagnostic features of this group, including the posterior, setose extension of segment IX, the broad, shelf-like structure of sternum IX, including the flat, tooth-like, apical setae, and the complex, membranous lobes of the phallus. It appears artificially in couplet 26, separated from related species because it lacks the strongly downcurved, apically rugose dorsolateral processes indicated in couplet 2. The collection of additional specimens may prove that the absence of the dorsolateral process is an aberration; if so, the specimens would lead to *C. fluminensis* in the key presented above.

Segment IX without mesolateral setae or setose protuberances (HR 3A, B); inferior appendage, in lateral view, subtriangular (HR 3A) ................................................................................................................................. ***Contulma meloi***[2]

27(26). Apex of phallus with large, apicoventral spines (HF 97–98); parameres absent ...................................................................................... ***Contulma talamanca***

Apex of phallus without such spines; parameres present, long, slender (HF 38, 102) ................................................................................................................ 28

28(27). Segment IX with patch of long, stout, spine-like mesolateral setae (HF 100); phallicata with parameres and lateral flanges (HF 102) ...................................... ......................................................................................................... ***Contulma tapanti***

Segment IX without patch of long, stout, spine-like mesolateral setae, although smaller setae may be present (HF 36); phallicata with parameres only, lateral flanges absent (HF 38) ....................................................... ***Contulma caldensis***

## DISCUSSION

*Contulma* species appear to be highly regionally endemic (*Holzenthal & Flint, 1995*; *Holzenthal & Robertson, 2006*). We observed this pattern in our collections in Ecuador. Patterns of high endemicity in macroinvertebrate benthic larvae are well known in high altitude glacial streams; these habitats are highly threatened by climate change (*Jacobsen et al., 2012*). On the other hand, we added records to the Ecuadorian fauna of species that were previously known only from Colombia, demonstrating that some species may have spread across the Andes. Despite this distribution pattern, aquatic insect species, especially mayflies, have shown genetic isolation in the various ranges of the Ecuadorian Andes, even with relatively short distances between populations and with recent volcanic eruption history (*Finn, Encalada & Hampel, 2016*). No similar study has yet been performed with any Andean caddisfly species, but we expect a similar pattern of isolation among populations. This isolation could increase by the intense land use changes and water pollution occurring throughout the inter-Andean valleys (*Ríos-Touma, Acosta & Prat, 2014*) potentially preventing dispersal among mountain populations and also local extinctions.

## ACKNOWLEDGEMENTS

We thank Dr. Desi Robertson, Dr. Brian Armitage, Dr. Tatiana I. Arefina-Armitage, and an anonymous reviewer for their constructive reviews of the manuscript.

### Funding

This study was supported by Universidad de Las Américas project AMB.BRT.17.005 "Diversidad y Distribución de Trichoptera de Ecuador" and Minnesota Agricultural Experiment Station projects MIN17-017 and 17-029. Ernesto Rázuri-Gonzales was funded through a doctoral fellowship from Cienciactiva, Consejo Nacional de Ciencia, Tecnología,

e Innovación Tecnológica Perú (contract 277-2015- FONCECYT). The funders had no role in study design, data collection and analysis, decision to publish, or preparation of the manuscript.

## Grant Disclosures

The following grant information was disclosed by the authors:

Universidad de Las Américas: AMB.BRT.17.005.

Minnesota Agricultural Experiment Station: MIN17-017, 17-029.

Cienciactiva, Consejo Nacional de Ciencia, Tecnología e Innovación Tecnológica Perú: 277-2015- FONCECYT.

## Competing Interests

The authors declare there are no competing interests.

## Author Contributions

- Ralph W. Holzenthal and Ernesto Rázuri-Gonzales conceived and designed the experiments, performed the experiments, analyzed the data, contributed reagents/materials/analysis tools, wrote the paper, prepared figures and/or tables, reviewed drafts of the paper.
- Blanca Ríos-Touma conceived and designed the experiments, performed the experiments, analyzed the data, contributed reagents/materials/analysis tools, wrote the paper, reviewed drafts of the paper, obtained collecting and export permits.

## Field Study Permissions

The following information was supplied relating to field study approvals (i.e., approving body and any reference numbers):

This study was performed under the Environmental Ministry of Ecuador study permits 36-2010-IC-FLO/FAU-DPA-MA and 005-15-IC-FAU-FLO-DNB/MA.

## Data Availability

Specimens forming the basis of the new species descriptions are deposited in the appropriate country institutions: Museo Ecuatoriano de Ciencias Naturales, Quito, Ecuador (MECN); and University of Minnesota Insect Collection, St. Paul, Minnesota, USA (UMSP).

## New Species Registration

The following information was supplied regarding the registration of a newly described species:

Publication LSID:

urn:lsid:zoobank.org:pub:54BC56DC-5CC1-4DA0-82E4-AF69599C2F5D

*Contulma lina*, new species

LSID urn:lsid:zoobank.org:act:310DDC54-0008-4535-A385-BD71BB630F21

*Contulma quito*, new species

LSID urn:lsid:zoobank.org:act:97969526-CF58-48A8-8548-8817326D04A0

*Contulma sangay*, new species

LSID urn:lsid:zoobank.org:act:38C17810-F31C-45A5-BFAC-3D3C5E68AD47.

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
