# Peer review of "New species of the endemic Neotropical caddisfly genus Contulma from the Andes of Ecuador (Trichoptera: Anomalopsychidae)"

_PeerJ, doi:10.7717/peerj.3967_

## Round 0.1 · original submission · Minor Revisions

As you will see, we received three very favorable reviews of your work. I believe that careful attention to the concerns and suggestions of all three reviewers will greatly improve your work and make it suitable for publication in PeerJ. In particular, please take a close look at the suggestions by Reviewer 3, as well as the marked-up files provided by reviewer 1 and reviewer 2. I very much look forward to receiving your revised manuscript, and best of luck with the revision.

-joe

Reviewer 1 ·

Basic reporting

The English used throughout the manuscript is sound and there are only a few minor typing mistakes. The literature is up to date and all important references are cited in the text. I suggest the addition of a few authorship references along the text. The manuscript has a coherent structure with no misplacement of text in wrong sections. The authors correctly present background information for the reader in the Introduction section and state their hypothesis. The results are new and relevant to science and in accordance with the issues stated in the introduction. The figures are well drawn and present clear information of the necessary issues.

Experimental design

The research presented is within the aims and scope of PeerJ. The methods used by the authors are those mostly used by the entomologists and suit correctly the aim of the manuscript. The methods are described in sufficient detail and references correctly cited.
The research question helps filling a biodiversity gap extant in caddisfly studies, especially in the still poorly known Andean area.

Validity of the findings

The species presented by the authors are undoubtedly new to science and are presented in a clear way with sufficient illustrations to support their asserts. The additional distributional data the authors provide are meaningful and help us to better understand the biodiversity and distribution of organisms. The authors propose a species synonym, which seems to be correctly stated based on the knowledge and material available. The argumentation they provide for the synonym is logic and the evidences provided are sufficient to lead to this conclusion.

Additional comments

The authors present three new species of the high altitude caddisfly genus Contulma. Based on the descriptions and illustrations provided, the species are definitely new and represent a significant contribution to caddisfly knowledge. They also present new records and an identification key of males of Contulma, which is a relevant contribution to science, especially taxonomy and ecology of aquatic insects. The manuscript is well written and contains only some minor issues that should be addressed, such as citation of figures and rearrangement of a few sentences. All those issues are marked in an annotated PDF.

Annotated reviews are not available for download in order to protect the identity of reviewers who chose to remain anonymous.

·

Basic reporting

This paper is well-written. The figures are very good, and the literature matches with the call-outs in the text. A good contribution to our knowledge of this genus. Some minor corrections are suggested in the PDF.

Experimental design

This paper falls within the Aims and Scope of the journal and meets all other stated objectives.

Validity of the findings

no comment

Additional comments

Please see yellow highlighted areas for minor suggested improvements. A table with all of the known species, their authorities, and country distribution would improve the overall value of this paper. This manuscript was also reviewed by Tatiana I. Arefina-Armitage.

·

Basic reporting

The paper is clear and well written. The authors present an excellent overview of the current state of Trichoptera taxonomy in the Neotropics and ample background information on the distribution, habitat, and previous work of the genus. The illustrations are superb and expertly done. Relevant collection, specimen, and database information are provided.

Experimental design

This original taxonomic research is within the scope of PeerJ and fills an important knowledge gap for Neotropical taxonomy. The work meets the highest technical and ethical standards in taxonomy. The methods are described in great detail and in such as way to facilitate replication by another investigator.

Validity of the findings

The superb illustrations, detailed diagnoses and descriptions, and identification key will prove to be valuable resources for future workers of this group. The authors present ample evidence for the designation of these new species. The authors explain their rationale for the synonymization of C. duffi with C. penai.

Additional comments

Thank you for the opportunity to review this manuscript.

This original taxonomic research is within the scope of the journal and appropriate for publication in PeerJ. The importance of describing rare, highly endemic new species, such as these new Contulma from the Andean region of Ecuador, cannot be overstated. This work is an excellent contribution to our overall knowledge of Trichoptera taxonomy and distribution, and provides a rare glimpse into the remarkable diversity of these little known organisms. Climate change, land-use changes, and other threats to high altitude glacial streams and inter-Andean valleys all underscore the critical need to document this diversity.

The paper is clear and well written. The authors present an excellent overview of the current state of Trichoptera taxonomy in the Neotropics and ample background information on the distribution, habitat, and previous work of the genus. Methods are clearly described, and include valuable information regarding databases and adherence to ICZN standards. The illustrations are superb and expertly done. The diagnoses and descriptions are detailed and clear. The synonymization of C. duffi with C. penai is well explained and justified. The inclusion of references to previously published illustrations, will make the identification key an especially valuable resource for future workers. I was unable to test the key using actual specimens, but based on previously published illustrations, it worked perfectly.

I offer a few very minor changes below as suggestions for improvement. I have also listed some very minor edits (mostly typographical errors). All of these are meant as suggestions to improve the paper and should be left to the authors as to whether or not they are needed.

1.) Please check for consistency in formatting for in-text citations. For example, on line 33, Holzenthal 1988 is written without a comma, however on line 48 Holzenthal and Río-Touma, 2012, the comma is included. I leave it to the PeerJ editors to confirm the proper final formatting.
2.) Lines 45-49: The authors explain that these species are rarely collected using standard light trap techniques and that most specimens are hand netted during the day or collected by Malaise traps, especially at high altitude. They go on to state that infrequency of collection does not equate to rarity in nature. Although not explicitly stated, the authors seem to imply that this may be a reflection of minimal collection efforts or perhaps low temperatures. Although admittedly speculative, perhaps a sentence to this effect would help to clarify.
3.) Line 97: Author & year not listed for first mention of C. paluguillensis.
4.) Lines 99-100: First mention of species listed do not cite authors & years. Are these also described by Holzenthal and Flint 1995?
5.) Line 194: Author & year not listed for first mention of C. lanceolate.
6.) While not critical to the paper, readers might appreciate a short recap of what is known regarding the phylogeny and species relationships of the genus. The authors cite Holzenthal and Flint’s (1995) comprehensive revision of Contulma, yet they make no mention of species groups, although they are somewhat alluded to in some of the diagnoses (e.g., lines 99-101, 152-153, Footnote 2 of key). I recognize a phylogenetic analysis is certainly outside the scope of this paper, but a broader discussion of morphology, species relationships, and how these new species might be placed, would add valuable context.
7.) Line 203: There is an extra “s” in the word process.
8.) Footnote 1 of Key: Robertson is incorrectly spelled.
9.) Line 249: posteromesal is incorrectly spelled.
10.) Line 384: setose protuberances (needs space).
11.) It is unclear why each of the figure legends begins with “Catalog of the Neotropical Trichoptera (Caddisflies)”. Typographical error?

In summary, I recommend this manuscript for publication with very minor revisions. It is an excellent work!

---

## Round 0.2 · accepted · Accept

Dear Dr. Holzenthal and coauthors:

Thank you for resubmitting your work to PeerJ. I am happy to inform you that your revision is now suitable for publication, so big congratulations! This is a very important work for the Trichoptera field, and will have an impact on the field of taxonomy in general. Regarding the table that you find redundant, I'll allow this to be omitted ONLY if the reference you refer to is open access. If the information in this previous work is not open access, then please provide the table in either the text of this manuscript, or as a supplement. Regarding your concerns with symbols, numbers, etc., I have forwarded these concerns to the production team, and if there is a problem, they will contact you directly. Thanks for drawing attention to these important matters. Again, congratulations, and thank you for submitting your revision and for choosing to publish with PeerJ. Well done to all!

-joe